# Phosphoproteins Involved in the Inhibition of Apoptosis and in Cell Survival in the Leiomyoma

**DOI:** 10.3390/jcm8050691

**Published:** 2019-05-16

**Authors:** Blendi Ura, Lorenzo Monasta, Giorgio Arrigoni, Ilaria Battisti, Danilo Licastro, Giovanni Di Lorenzo, Federico Romano, Michelangelo Aloisio, Isabel Peterlunger, Guglielmo Stabile, Federica Scrimin, Giuseppe Ricci

**Affiliations:** 1Institute for Maternal and Child Health–IRCCS “Burlo Garofolo”, 65/1 Via dell’Istria, I-34137 Trieste, Italy; lorenzo.monasta@burlo.trieste.it (L.M.); giovanni.dilorenzo@burlo.trieste.it (G.D.L.); federico.romano@burlo.trieste.it (F.R.); michelangeloaloisio@libero.it (M.A.); isabel.peterlunger@libero.it (I.P.); guglielmost@gmail.com (G.S.); federica.scrimin@burlo.trieste.it (F.S.); giuseppe.ricci@burlo.trieste.it (G.R.); 2Department of Biomedical Sciences, University of Padova, 35121 Padova, Italy; giorgio.arrigoni@unipd.it; 3Proteomics Center, University of Padova and Azienda Ospedaliera di Padova, I-35129 Padova, Italy; ilaria.battisti@studenti.unipd.it; 4CBM scrl, Area Science Park, I-34149 Trieste, Italy; danilo.licasto@cbm.fvg.it; 5Department of Medical, Surgery and Health Sciences, University of Trieste, I-34149 Trieste, Italy

**Keywords:** phosphoproteomics, leiomyoma, 2-DE, myometrium, mass spectrometry

## Abstract

Uterine leiomyomas are benign smooth muscle cell tumors originating from the myometrium. In this study we focus on leiomyoma and normal myometrium phosphoproteome, to identify differentially phosphorylated proteins involved in tumorigenic signaling pathways, and in anti-apoptotic processes and cell survival. We obtained paired tissue samples of seven leiomyomas and adjacent myometria and analyzed the phosphoproteome by two-dimensional gel electrophoresis (2-DE) combined with immobilized metal affinity chromatography (IMAC) and Pro-Q Diamond phosphoprotein gel stain. We used mass spectrometry for protein identification and Western blotting for 2-DE data validation. Quantities of 33 proteins enriched by the IMAC approach were significantly different in the leiomyoma if compared to the myometrium. Bioinformatic analysis revealed ten tumorigenic signaling pathways and four phosphoproteins involved in both the inhibition of apoptosis and cell survival. Our study highlights the involvement of the phosphoproteome in leiomyoma growth. Further studies are needed to understand the role of phosphorylation in leiomyoma. Our data shed light on mechanisms that still need to be ascertained, but could open the path to a new class of drugs that not only can block the growth, but could also lead to a significant reduction in tumor size.

## 1. Introduction

Uterine leiomyomas are benign smooth muscle cell tumors originating from the myometrium. These tumors can occur in 70–80% of women and are the first gynecological reason why women undergo hysterectomy [1]. They are responsible for severe health problems, including pain, infertility, repetitive pregnancy loss and heavy uterine bleeding [2]. To date, little is known about the pathogenesis of this tumor, and this leads to the lack of a valid medical treatment. Leiomyomas are characterized by a hyperproduction of extracellular matrix (ECM) [3], and by high interstitial fluid pressure (P_if_) [4].

Phosphorylation occurs in target amino acids, usually in serine, threonine, or tyrosine residues [5]. It is known that phosphorylation is the most relevant post-translational protein modification [6] and that altered phosphorylation in cellular pathways is strongly associated with tumors [7].

Hermon TL et al. showed that estrogen receptor α (ERα) is significantly more phosphorylated on serine in the leiomyoma, if compared to the myometrium, and is regulated by ERK1/2 leading to leiomyoma development [8,9]. Over-phosphorylation of receptor tyrosine kinases (RTKs), that regulate several cellular processes in tumors, like differentiation and proliferation, is associated with cell proliferation in the leiomyoma [10]. Signals of other cell membrane receptors, like transforming growth factor beta (TGF-β), activin, myostatin are transmitted to the nucleus through Smads proteins. The intracellular effectors of TGF-β signaling lead to the activation of Smad proteins by phosphorylation. These proteins are transferred to the nucleus, modulating gene expression [11]. In the leiomyoma, Smad2/3 is over phosphorylated if compared to the myometrium, and the use of GnRH-agonist treatment induces a reduction in phosphorylation of this protein [12]. Several studies have shown the dysregulation of the PI3K/Akt/mTOR pathway. Indeed, they find an increase in cyclin D_2_ and glycogen synthase kinase 3 (GSK3) and a decrease of phosphatidylinositol 3,4,5-trisphosphate 3-phosphatase (PTEN) leading to leiomyoma development [13]. There is evidence that the Ras/Raf/MEK/ERK pathway is dysregulated in the leiomyoma. Several studies have shown that different kinases, such as ERK and RTK, are altered in this pathway and this may be related to the disease [14,15].

For the reasons described above, the identification of changes in protein phosphorylation levels in the leiomyoma may be useful for the understanding of the tumor physiopathology. Several proteomic studies have been conducted and studies have identified proteins involved in leiomyoma pathology [16,17].

The objective of our study was to create a phosphoproteomic profile by using Pro-Q Diamond and immobilized metal affinity chromatography (IMAC) of leiomyoma and myometrial tissues, for the identification of differentially phosphorylated proteins involved in tumorigenic signaling pathways and in anti-apoptotic and cell survival processes.

## 2. Materials and Methods

### 2.1. General

Tissues samples were obtained from seven premenopausal patients who underwent hysterectomy for symptomatic uterine leiomyomas. All procedures conformed to the Declaration of Helsinki and were approved by the Review Board of the Institute for Maternal and Child Health–IRCCS “Burlo Garofolo” (Trieste, Italy). All subjects involved signed a written informed consent. The median age of patients was 45 years, with a minimum of 36 and a maximum of 48 years. 

Oncologic patients; Human immunodeficiency virus (HIV), Hepatitis B virus (HBV), Hepatitis C virus (HCV) seropositive patients; and patients with adenomyosis were excluded from the study. The patients had not received hormonal therapy in the three months before surgery.

### 2.2. Tissue Samples

Two samples were collected from each patient: one from the central area of the leiomyoma and one from the unaffected myometrium situated more than 2 cm away from the leiomyoma capsule. All leiomyomas were confirmed histologically as benign ordinary leiomyomas. All leiomyomas were subserosal/intramural with dimensions ranging from 4 to 6 cm. Samples were stored at −80 °C until proteomic analysis was performed.

### 2.3. Phosphoprotein Isolation, Pro-Q Diamond and 2-DE 

One hundred mg of myometrium and leiomyoma tissue from seven patients were used for phosphoprotein isolation by Phosphoprotein Enrichment Kit (Pierce). Tissues were homogenized in (1% NP-40, 50 mM Tris-HCl (pH 8.0), NaCl 150 mM) with Phosphatase Inhibitor Cocktail Set II 1× (Millipore, Burlington, VT, USA) and 2 mM phenylmethylsulfonyl fluoride (PMSF), 1 mM benzamidine. The concentration of the supernatant was determined by Bradford. Tissue homogenates were then diluted to 0.4 mg/ml in lysis buffer provided by the Phosphoprotein Enrichment Kit. Six ml of final samples were used for isolation of the phosphoprotein according to the manufacturer’s instructions.

2-DE was performed as previously described [18]. For 2-DE analysis, 250 μg of proteins from each sample (obtained after phosphoproteins enrichment, as specified above) were denatured in 300 μl of dissolution buffer (7 M urea, 2 M thiourea, 4% 3-((3-cholamidopropyl)dimethylammonio)-1-propanesulfonate, 40 mM Tris, 65 mM dithiothreitol (DTT), and 0.24% Bio-Lyte 3/10 (Bio-Rad Laboratories, Inc., Hercules, CA, USA) and a trace of bromophenol blue. ReadyStrip™ pH 4–7 18-cm immobilized pH gradient (IPG) strips (Bio-Rad Laboratories, Inc., Hercules, CA, USA) were rehydrated in a dissolution buffer at 50 V for 12 h at 20 °C, and isoelectric focusing (IEF) was performed in a PROTEAN IEF Cell (Bio-Rad Laboratories, Inc., Hercules, CA, USA). After the IEF, serial incubations were performed: first, the IPG strips were equilibrated for 20 min in an equilibration buffer (6 M urea, 2% SDS, 50 mM Tris-HCl (pH 8.8), 30% glycerol and 1% DTT) and then in another equilibration buffer containing 4% iodoacetamide instead of DTT. For the second dimension, the equilibrated IPG strips were transferred to a 12% polyacrylamide gel.

After electrophoresis, gels were fixed in 40% methanol and 10% acetic acid for 1 h, and then stained for 16h with SYPRO Ruby. 2-DE gels were scanned with a Molecular Imager PharosFX System. Double experimental replicates were performed per sample.

For the experiment with Pro-Q Diamond dye (Thermo Life Technologies, Waltham, MA, USA), seven patients (the same patients used for phosphoproteome enrichment) were used. One thousand μg of proteins were used for 2-DE as described above. Gels were first stained with Pro-Q Diamond dye according to manufacturer’s instructions, washed with water, and scanned with Pharos FX Plus Molecular Imager (Bio-Rad) to visualize the phosphoproteome. Gels were stained for 16 h with SYPRO Ruby to reveal the total proteome. Double experimental replicates were performed per each sample. For all gels, molecular weights were determined by comparison with Precision Plus Protein Pre-stained Standards (Bio-Rad Laboratories, Inc., Hercules, CA, USA), covering a range from 10 to 250 kDa and analyzed using the Proteomweaver 4.0 software (both from Bio-Rad Laboratories, Inc., Hercules, CA, USA).

### 2.4. Quantification of Spot Levels

2-DE image analysis was performed using the Proteomweaver 4.0 software. The analysis process was carried out by matching all gels from seven myometria and seven leiomyomas. The Proteomweaver 4.0 algorithm matched all of the gels to find quantitative differences. Differences were considered significant when the ratio of the mean percentage relative volume (%*V*) (%*V* = *V*(single spot)/*V*(total spot)) was ±1.5-fold and satisfied the non-parametric Wilcoxon test (*P* < 0.05). For Pro-Q Diamond gel stain, the ratios of relative protein abundance values between the myometrium and the leiomyoma were calculated. Ratios ≥ 1.5 and ≤ 0.6 were considered as significantly different. The relative protein abundance of phosphoproteins (P) was calculated in the Pro-Q Diamond images and in the SYPRO Ruby images, as previously described by Wang and colleagues [19].

### 2.5. Trypsin Digestion and MS Analysis

Spots from 2-DE were digested and analyzed by mass spectrometry, as described by Ura et al. [20].

After excision from 2-DE gels, the spots were washed four times with 50 mM NH_4_HCO_3_ and acetonitrile (ACN; Sigma-Aldrich, St. Louis, MO, USA) alternatively, and dried under vacuum in a SpeedVac system. For gel spot digestion, three microliters of 12.5 ng/µL sequencing grade modified trypsin (Promega, Madison, WI, USA) in 50 mM NH_4_HCO_3_ were added, and samples were digested overnight at 37 °C. Peptides extraction was made by three changes of 50% ACN/0.1% formic acid (FA; Fluka, Ammerbuch, Germany), peptide mixtures were dried under vacuum and stored at −20 °C, until mass spectrometry (MS) analysis was performed. 

Samples were dissolved in 10 µL of 5% ACN/0.1% FA and 5 microliters of each sample were analyzed by LC-MS/MS on a 6520 Q-TOF mass spectrometer (Agilent Technologies, Santa Clara, CA, USA) coupled to a chip-based chromatographic interface. A Large Capacity Chip was used and peptides were separated in the C18 nano-column (150 mm × 75 μm ID) at a flow rate of 0.3 μL/min. H_2_O/FA 0.1% and ACN/FA 0.1% were used as eluents A and B, respectively. Peptides were separated with a linear gradient of eluent B from 5% to 50% in 15 min and analyzed with a data dependent mode acquisition: for each MS scan, 6 MS/MS spectra were acquired for the most intense ions. Scan speeds were 3 MS spectra/sec and 3 MS/MS spectra/sec. 

Raw data files were converted into Mascot Generic Format (MGF) files with MassHunter Qualitative Analysis Software version B.03.01 (Agilent Technologies, Santa Clara, CA, USA) and searched with Mascot Search Engine (version 2.2.4, Matrix Science, London, UK) through the Proteome Discoverer Software interface (version 1.4, Thermo Fisher Scientific, Waltham, MA, USA). Spectra were searched against the human section of the Uniprot database (version July 2018, 95,057 sequences) using the following parameters: enzyme specificity was set to trypsin with 1 missed cleavage allowed, precursor and fragment ions tolerance were 20 ppm and 0.05 Da, respectively. Carbamidomethylcysteine and oxidation of methionine were set as fixed modification and variable modification, respectively. MS/MS spectra containing less than 5 peaks or with a total ion count lower than 50 were discarded. The algorithm Percolator was used to assess the False Discovery Rate (FDR) thanks to a concomitant search against the corresponding randomized database. Proteins were considered as positive hits if for each protein at least 2 unique peptides were identified with high confidence (FDR < 0.01%). For some protein spots that did not return any significant hit, a Peptide Mass Fingerprint (PMF) was also performed with Mascot. All identified proteins were verified to have phosphorylated residues in PhosphoSitePlus database (www.phosphosite.org).

### 2.6. Western Blotting

Phosphoprotein extracts (20 μg) from IMAC columns used for 2-DE were separated by 12% and then transferred to a nitrocellulose membrane. The Western blotting procedure for phosphoproteins was conducted as previously described [21]. The membrane was blocked by treatment with 5% BSA in TBS-tween 20. After BSA saturation the membrane was incubated overnight at 4 °C with 1:200 diluted primary rabbit polyclonal antibody against Endoplasmic reticulum chaperone BiP (HSPA5), with 1:300 diluted primary rabbit polyclonal antibody against heat shock protein beta-1 (HSPB1), and 1:800 diluted primary rabbit polyclonal antibody against Vinculin (VINC). The membrane was washed three times in TBST for 10 min, and then incubated for 90 min at 4 °C with a horseradish peroxidase-conjugated anti-rabbit immunoglobulin G antibody (Sigma-Aldrich, St. Louis, MO, USA; Merck KGaA, Burlington, VT, USA) at 1:3.000 dilution. Protein expression was visualized by chemiluminescence (SuperSignal West Pico Chemiluminescent Substrate; Thermo Fisher Scientific, Inc., Waltham, MA, USA), and the intensity of the signals was quantified by VersaDoc Imaging System (Bio-Rad Laboratories, Inc.). The intensities of the immunostained bands were normalized with the protein intensities measured by Red Ponceau (Sigma-Aldrich, St. Louis, MO, USA; Merck KGaA, Burlington, VT, USA) from the same blot.

### 2.7. Ingenuity Pathway (IPA) and PANTHER Analysis

Differentially enriched proteins identified by MS in the leiomyoma were analyzed by IPA (Qiagen GmbH, Hilden, Germany). In IPA, we considered *P* < 0.05 as a statistically significant value. Selected genes were used to generate bio-functions. For the filter summary, only associations where confidence was high (predicted) or that had been experimentally observed were considered. These proteins were analyzed by PANTHER 11.0 (Protein Analysis Through Evolutionary Relationships; http://www.pantherdb.org) and Gene Ontology (http://amigo.geneontology.org/rte). Proteins were then classified according to their involvement in biological processes, molecular function, protein class, and pathways. Since the majority of the proteins identified participated in multiple processes, only the most relevant ones were reported.

### 2.8. Statistical Analysis

Statistical analyses were carried out with the non-parametric Wilcoxon signed-rank test for matched samples for both 2-DE and Western blot data. *P* < 0.05 was considered to indicate a statistically significant difference. All analyses were conducted with Stata/IC 14.1 for Windows (StataCorp LP, College Station, TX, USA).

## 3. Results

### 3.1. Identification of Differentially Phosphorylated Proteins with Pro-Q Diamond Gel Stain

After electrophoresis, gels were stained with Pro-Q Diamond for phosphoproteins (P) and with SYPRO Ruby for total proteins (T). Correlation analysis of gel-pairs performed well, with an average matching efficiency of approximately 80%. As shown in Figure 1, ten protein spots belonging to seven different putative phosphoproteins were found to be with a significant different abundance in leiomyoma samples if compared to the myometrium. The spot volume of two of these proteins [corresponding to Apolipoprotein E (APOE) (spot6) and 60S acidic ribosomal protein P2 (RPLP2) (spot7)] was significantly higher (>1.5-fold) (Table 1) in the leiomyoma, while for five proteins (Caveolae-associated protein 1 (CAVN1) (spot1,2,3,4), Heterogeneous nuclear ribonucleoproteins C1/C2 (HNRNPC) (spot5), Caveolae-associated protein 1 (PTRF) (spot8), Apolipoprotein D (APOD) (spot9) and Myosin light chain kinase, smooth muscle (MYLK) (spot10)) the relative spot volumes were significantly lower (<0.6-fold, Figure 1). 

### 3.2. Enrichment of Immobilized Metal Affinity Chromatography (IMAC)

The next step for the identification of differentially phosphorylated proteins in the leiomyoma was the using of IMAC. This method, although not completely specific, allows for an efficient enrichment of phosphoproteins. By using 2-DE fluorescence, we compared the enriched phosphoproteome of the myometrial and leiomyoma tissues. An average of 1800 spots was detected on gels for both types of enriched phosphoproteomes. Correlation of gel-pairs performed with an average matching efficiency of ~80%. The analysis revealed 35 protein spots with a significant different abundance (Figure 2) in the leiomyoma compared to myometrium. We considered only spots supposedly corresponding to phosphoproteins, with fold change in %V ≥1.5 or ≤0.6 in intensity and that were statistically significant (*P* < 0.05). Of these spots, 34 were upregulated (>1.5-fold) while one was significantly downregulated (<0.6-fold) (Table 1). The 33 putative phosphoproteins were identified using a nano-LC-ESI-Q-TOF system and Mascot search engine, as described above.

### 3.3. Functional Analysis of the Leiomyoma Phosphoproteome

The identified proteins were divided into groups, based on the PANTHER classification system according to their biological processes, molecular function, protein class. In terms of biological processes (Appendix A), the proteins were grouped into four main categories: response to stimulus, cellular process, metabolic process, biological regulation. For molecular function (Appendix A), proteins were grouped into: catalytic activity, binding, molecular function regulator, and transporter activity. For protein class (Appendix A), proteins were grouped in five main categories: hydrolase, signaling molecule, oxidoreductase, enzyme modulator, and defense/immunity protein. For cellular component (Appendix A), proteins were grouped in four main categories: cell parts, organelle, protein-containing complex, extracellular region. For pathway analysis (Appendix A), proteins were grouped in 15 pathways. Ten of these (FAS signaling pathway, p38 MAPK pathway, VEGF signaling pathway, Cytoskeletal regulation by Rho GTPase, Integrin signaling pathway, Apoptosis signaling pathway, Angiogenesis, Gonadotropin-releasing hormone receptor pathway, Ubiquitin-proteasome pathway, Inflammation mediated by chemokine and cytokine signaling pathway) are tumorigenic signaling pathways.

Proteins differentially enriched by IMAC in the leiomyoma compared to the myometrium were used in the core analysis with IPA software. The top networks in which these proteins are involved correspond to 1) Apoptosis (Appendix AA.) and 2) Cell survival (Appendix AB.). In the apoptosis network, 18 putative phosphoproteins are involved. Of these, CFH (spot36), DYNC (spot28), HSPA5 (spot29), HSPB1 (spot14), HSPD1 (spot39), PDRX2 (spot12), UBA1 (spot33,34), A2M (spot37) are involved in apoptosis inhibition, while HSPA5, HSPB1, HSPD1, LMNA (spot20), P4HB (spot41,42), PRDX2, PSMB4 (spot13) are involved in cell survival. HSPA5, HSPB1, HSPD1 and PDRX2 are involved in both inhibition of apoptosis and cell survival. 

### 3.4. Immunohistochemical Study of Altered Proteins

In this study, we proceeded to verify by immunoblot the alteration of three putative phosphorylated proteins: HSPA5, HSPB1, and VINC. After IMAC enrichment, the abundance of these proteins in five leiomyomas was compared to the matched normal myometrial tissue samples (previously used in 2-DE analysis) by Western blot analysis (Figure 3). The five patients shown in the figure are representative of the total seven patients included in the study, based on both 2-DE and Western blotting phosphorylation trend of HSPB1, HSPA5, and VINC (higher in leiomyoma compared to myometrium). We selected HSPB1 and HSP5A because, based on our bioinformatic analysis, these two proteins are associate to the inhibition of apoptosis and cell proliferation. We selected VINC because the antibody is commercially available.

## 4. Discussion

Several studies have shown that the phosphorylation of certain proteins like ER, AKT, RTK, MAPK, MEK, and GSK3 lead to increased leiomyoma cell proliferation [8,9,22]. For the first time, we conducted a proteomic study on leiomyoma and normal myometrium to identify putative phosphoproteins and the pathways in which these proteins are involved and that are altered in the tumor as compared to the normal tissue. By using a combination of IMAC enrichment, 2-DE, Diamond Pro-Q staining, and mass spectrometry, we identified 32 putative phosphoproteins differentially phosphorylated/expressed in the leiomyoma as compared to the myometrium (Table 1). We identified several tumor-related pathways in which these proteins are involved and that could be implicated in the modulation of leiomyoma growth. According to the data reported in the PhosphoSitePlus database, all identified proteins were already reported to be phosphorylated in vivo, suggesting that the enrichment procedure was effective in isolating mainly phosphorylated proteins. The altered abundance or phosphorylation level of some of these proteins was confirmed by an independent method (IMAC enrichment, 2-DE, and Western blotting). However, we did not try to identify the phosphorylation site(s) that could be altered in these proteins and therefore our validation could not be performed using antibodies against specific phosphorylated sites. Nonetheless, we could confirm a significant difference of these proteins in tumor vs. normal tissue, either due to an overall altered abundance of the proteins or to an altered phosphorylation level. 

Jamaluddin MFB et al. [14] conducted a proteomic study for the characterization of fibroid ECM proteins. They identified several proteins up regulated in the leiomyoma: POSTN, TNC, COL3A1, COL24A1, and ASPN. Another proteomic study conducted by Liu Y et al. [23] identified TRADD as a potential biomarker in human uterine leiomyoma. 

The Fas receptor (CD95) mediates apoptotic signaling initiated by interaction with surface expressed FasL on other cells [24]. Mutations in the Fas signaling pathway can result in cell hyperproliferation [25]. Once the complex Fas-FasL has been created, the apoptotic program should be activated via recruitment of signaling adaptor molecules to the Fas receptor [26].

Lamins constitute a class of intermediate filaments, which, during mitosis, are dissociated by phosphorylation-dependent mechanisms [27]. To date, no data regarding the activity of the phosphorylated form of Lamin A have been reported. According to Foster CR and colleagues, increased mobility of phosphorylated Lamin A could be one way to promote nuclear deformability and cell migration [28].

The p38 MAPK pathway is regulated by stress, cytokines and mediates a wide variety of cellular behaviors, such as cell differentiation, proliferation, migration, invasion, and cell death [29].

The VEGF pathway represents the signal transduction cascade, initiated after this protein binds to its receptor VEGFR-2, leading to survival, increased proliferation, and migration of blood endothelial cells required for angiogenesis [30]. In the leiomyoma, VEGF is overexpressed and leads to tumor growth in xenografts mice [31]. In this study, we identified four putative phosphoproteins involved in anti-apoptotic and cell survival processes, events that promote cell growth.

HSPB1 plays a role in stress resistance and actin organization. In response to environmental stress, the encoded protein translocates from the cytoplasm to the nucleus and functions as a molecular chaperone that promotes the correct folding of other proteins and plays an essential role in the differentiation of a wide variety of cell types [32]. This protein is correlated with poor clinical outcomes in several human cancers, promoting cancer cell proliferation and metastases, while protecting cancer cells from apoptosis [33]. In our previous study [34]. We predicted the phosphorylation of HSPB1 in the leiomyoma. In this study, we identified this protein as supposedly differentially phosphorylated in the leiomyoma, suggesting a possible role of phosphorylated HSPB1 in promoting cell survival in cancer [35].

HSPA5 is another putative phosphoprotein identified in this study, involved in cell survival and anti-apoptotic events. This protein plays a central role in regulating the unfolded protein response (UPR), is an obligatory component of autophagy in mammalian cells, and plays an important role in cellular adaptation and oncogenic survival [36,37]. This protein undergoes phosphorylation at specific serine and tyrosine residues, which influences their oligomerization into larger, multimeric aggregates [38]. Lim et al. suggest the possibility of a direct role of the phosphorylation of HSP5A in human breast cancer tissue [39].

HSPD1 is implicated in mitochondrial protein import and macromolecular assembly [40]. This chaperone promotes the refolding and proper assembly of unfolded polypeptides generated under stress conditions in the mitochondrial matrix [41]. The tyrosine phosphorylation of HSPD1 is essential for the maturation of ULBP2 (cell surface proteins that are present in transformed and stressed cells and ligands for NK cells) [42]. In cancer, the loss of ULBP2 leads to reduced NK cell-mediated cytotoxicity and cytokine production [43].

PRDX2 is the last of the identified putative phosphoproteins involved in anti-apoptotic and cell survival processes. PRDX2 is a member of the peroxiredoxin family of antioxidant enzymes, involved in redox regulation of the cell [44]. This enzyme influences several cellular processes involving proliferation, survival, and apoptosis, which suggests a possible role for Prdx2 in the maintenance of cancer cell [45]. Qu D et al. suggest that the Cdk5 phosphorylate PRDX2 [46]. This phosphorylation reduces PRDX2 activity thus increasing oxidative stress and cell proliferation [46]. We think that a similar mechanism may be present in the leiomyoma. Panther analysis revealed a significant enrichment of GO terms for several important processes, like metabolic processes and cellular biogenesis. Some of these metabolic processes have been previously described by our group [25]. pointing to a particular importance of metabolism in leiomyoma growth. In addition, our Panther analysis reveals that many proteins identified in our study are enzymes, are related to oxidative stress, and reside mostly inside the cell.

A scrutiny of the literature reveals that three kinases are known to be responsible for the phosphorylation of several proteins that we have identified in this study. In particular, MAPK, PKC, and Cdk5 were reported to phosphorylate HSPB1 [47], HSPA5 [48], HSPD1 [49], and PRDX2 [47]. Interestingly, both MAPK and PKC are known to be involved in leiomyoma development [9] and in general all three kinase are related with cancer cell progression and proliferation [47].

Several studies have been conducted to highlight the effects of various kinase inhibitors in the leiomyoma. Shushan A et al. [50] used the AG1478 an EGFR kinase blocker in leiomyoma cells. Leiomyoma cell growth is inhibited by AG1478, and is unaffected by the presence of physiological concentrations of estradiol and progesterone. In another paper, Shushan A et al. [51] evaluate the efficiency of genistein (a plant flavonoid) and the new protein tyrosine kinase inhibitor TKS050 in the inhibition of autophosphorylation of EGFR and downstream signal transduction events, including cell proliferation and cell cycle progression.

The ubiquitination is a poorly studied process in leiomyoma, even though it is known that the blocking of this process is related to tumor growth. An example is the switching of the ubiquitin/proteasome-dependent degradation of RXRα by phosphorylation in leiomyomas. These events may be responsible for the accumulation of RXRα and the consequent dysregulation of retinoic acid target genes [52].

UBA1 catalyzes the first step in ubiquitin conjugation to mark cellular proteins for degradation through the ubiquitin-proteasome system [53]. The role of this enzyme has been linked to DNA repair, for response to replication stress, cell cycle regulation, apoptosis, and cancer [54]. UBA1 inhibitors lead to an unfolded protein response and induces cell death in malignant cells over normal cells [55].

## 5. Conclusions

In conclusion, our study highlights the involvement of the phosphoproteome in leiomyoma growth. We identified several pathways associated with tumor development and phosphoproteins involved in the inhibition of apoptosis and cell proliferation. All these data show the existence of a relation between phosphorylation and leiomyoma development. Further studies are needed to understand the role of phosphorylation in leiomyoma growth. In our opinion, our data represent a step forward in the difficult understanding of the molecular mechanisms that lead to the formation and development of the leiomyoma.

## Figures and Tables

**Figure 1 jcm-08-00691-f001:**
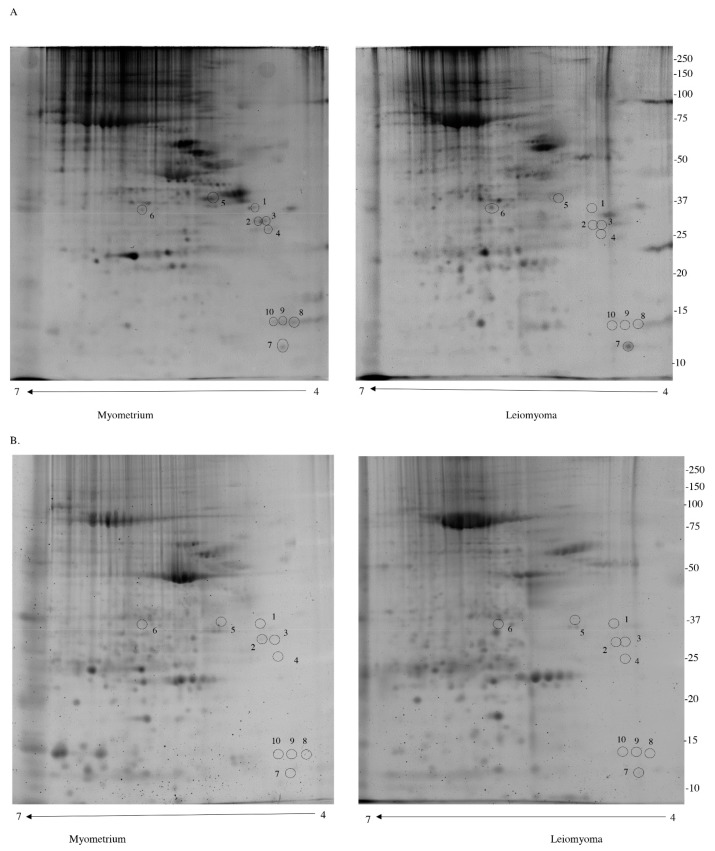
(**A.**) Images of gels stained with Pro-Q Diamond to detect phosphoproteins. (**B**) Images of gels stained with SYPRO Ruby for total proteins.

**Figure 2 jcm-08-00691-f002:**
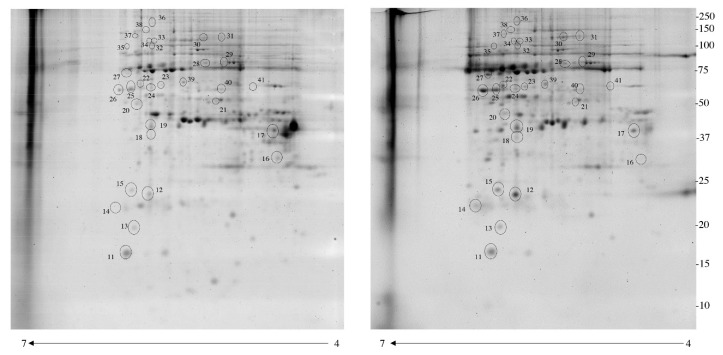
Two-dimensional electrophoresis map of normal myometrium and leiomyoma phosphoproteome enriched by IMAC columns. Immobilized pH gradient 4–7 strips were used for the first dimension and 12% polyacrylamide gel was used for the second dimension. The numbered circles indicate the differentially phosphorylated spots.

**Figure 3 jcm-08-00691-f003:**
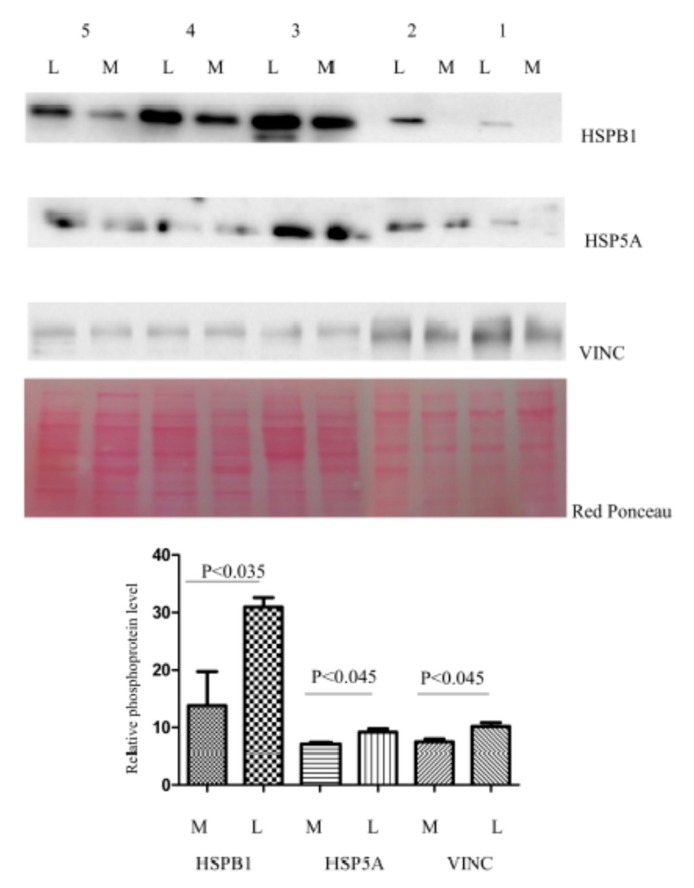
Western blot analysis was utilized to confirm the alteration of phosphorylation of proteins HSPB1, VINC and HSPA5 in paired myometrium (M) and leiomyoma (L). The intensity of immunostained bands was normalized against the total protein intensities measured from the same blot stained with Red Ponceau. Number 1-5 indicate the patients. The bar graph shows the relative expression (band density) of HSPB1, VINC and HSPA5 in the myometrium and the leiomyoma. Results are shown as a histogram (*P* < 0.05) and each bar represents mean ± standard deviation.

**Table 1 jcm-08-00691-t001:** List of putative phosphoproteins with a significant different abundance in the leiomyoma compared to myometrium, as identified by mass spectrometry.

Accession No	Spot No	Protein Description	Gene Symbol	Protein Score	Fold Change *	*P*-Value
P30101	22	Protein disulfide-isomerase A3	PDIA3	295	5.2	0.0277
H9KV75	25	Alpha-actinin-1	ACTN1	93	4.9	0.0431
P11021	29	78 kDa glucose-regulated protein	HSPA5	675	4.6	0.045
P05387	7	60S acidic ribosomal protein P2	RPLP2	121	4.5	0.0422
H0YJW3	32	Alpha-actinin-1 (Fragment)	ACTN1	141	3.8	0.0273
Q5JRR6	34	Ubiquitin-like modifier-activating enzyme 1	UBA1	64	3.8	0.0277
A0A087WU08	18	Haptoglobin	HP	159	3.4	0.0180
A0A087WU08	19	Haptoglobin	HP	50.8	3.3	0.0180
E9PFZ2	30	Ceruloplasmin	CP	154	3.2	0.0431
H7BZ94	41	Protein disulfide-isomerase	P4HB	272	3.2	0.0431
A0A0C4DGB6	26	Serum albumin	ALB	43	3	0.0220
H7C3T4	15	Peroxiredoxin-4	PRDX4	159	2.85	0.0273
Q5JRR6	33	Ubiquitin-like modifier-activating enzyme 1	UBA1	128	2.7	0.0273
P02790	27	Hemopexin	HPX	69	2.56	0.0431
Q3BDU5	20	Prelamin-A/C	LMNA	395	2.5	0.0277
H7BZ94	40	Protein disulfide-isomerase	P4HB	72	2.3	0.0277
P18206-2	38	Isoform 1 of Vinculin	VINC	198	2.25	0.0277
E9PFZ2	31	Ceruloplasmin	CP	237	2.25	0.0431
P30101	25	Protein disulfide-isomerase A3	PDIA3	808	2.23	0.0220
P10809	39	60 kDa heat shock protein	HSPD1	234	2.2	0.0431
P02649	6	Apolipoprotein E	APOE	230	2.15	0.0431
P01023	37	Alpha-2-macroglobulin	A2M	52	2.14	0.0431
P01023	12	Peroxiredoxin-2	PRDX2	64	2.1	0.0180
P04792	14	Heat shock protein beta-1	HSPB1	151	2	0.0277
P01024	17	Complement C3	CO3	171	2	0.0431
E9PN50	21	26S protease regulatory subunit 6A (Fragment)	PSMC3	191	2	0.0425
P30101	23	Protein disulfide-isomerase A3	PDIA3	79	2	0.0220
B7ZAR1	24	T-complex protein 1 subunit epsilon	CCT5	150	2	0.0277
P28070	13	Proteasome subunit beta type-4	PSMB4	168	1.75	0.0431
J3KRH2	11	Haptoglobin (Fragment)	HP	342	1.7	0.0178
P08603	36	Complement factor H	CFH	69	1.7	0.0273
Q13409-2	28	Isoform 2B of Cytoplasmic dynein 1 intermediate chain 2	DYNC1/2	85	1.51	0.0180
Q6NZI2	4	Caveolae-associated protein 1	CAV1	75	0.65	0.0277
Q6NZI2	2	Caveolae-associated protein 1	CAV1	161	0.43	0.03
Q6NZI2	3	Caveolae-associated protein 1	CAV1	127	0.37	0.0180
G3V5V7	5	Heterogeneous nuclear ribonucleoproteins C1/C2 (Fragment)	HNRNPC	69	0.29	0.0180
Q6NZI2-3	8	Isoform 3 of Polymerase I and transcript release factor	PTRF	238	0.11	0.0422
C9JF17	9	Apolipoprotein D	APOD	103	0.1	0.0422
Q15746-10	10	Isoform 8 of Myosin light chain kinase, smooth muscle	MYLK	51	0.08	0.03
P17661	16	Desmin	DES	787	0.06	0.0180
Q6NZI2	1	Caveolae-associated protein 1	CAV1	63	0.03	0.0178

* Fold change was defined as the ratio of the mean %V according to the formula %V = Vsingle spot/Vtotal spot of cases vs. controls.

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
