# Peer review of "Phosphoproteins Involved in the Inhibition of Apoptosis and in Cell Survival in the Leiomyoma"

_jcm, 2019, doi:10.3390/jcm8050691_

Reviewer 1 Report

 The manuscript by Ura et al entitled “Phosphoproteins involved in the tumorigenic signaling pathway, in the inhibition of apoptosis and in cell survival, identified through the phosphoproteomic analysis of the leiomyoma” focused on phosphoproteome in leiomyoma and adjacent myometrium.  The authors isolated total proteins from paired samples of leiomyoma and the adjacent myometrial tissues in 7 patients, enriched phosphoproteins by immobilized metal affinity chromatography, separated the proteins by Two-Dimensional gel electrophoresis, visualized the phosphorylated proteins with Pro-Q Diamond staining, identified differentially phosphorylated proteins by mass spectrometry. A total of 33 (?) phosphoproteins are identified to be differentially expressed. By using ingenuity pathway analysis, the authors identify the differentially expressed phosphoproteins involving in tumorigenic signaling pathways, anti-apoptotic processes, and cell survival.

Major concerns:

A.     The protease inhibitors used in the sample preparation are major concerns. PMSF is a serine hydrolase inactivator and benzamidine is an inhibitor of trypsin, trypsin-like, and serine proteases. Inhibitors for cysteine protease, metalloprotease, calpain, and other proteases are not applied in the procedure. This may result in degradation of the phosphoproteins into small fragments that cannot be separated by 2-DE. Families of phosphoproteins or signal transduction pathways may be missed that could be physiologically important and alter the conclusion of the study.

B.     The authors compared the phosphoproteins between paired diseased and adjacent normal tissues from same patients. The change of abundance of protein spots is not so substantial, Ratios ≥ 1.5 and ≤ 0.6 are considered as significantly different. Comparison between diseased patients with “normal” patient controls may be more meaningful.

C.     The biographic data of the patients are limited and clinic data are missing. The diagnosis is not clear. The histopathology of the diseased or adjacent tissues are not shown. The distance from diseased tissue to adjacent normal tissue is not described.

­­­­­­­­­­­­­­

D.    In Figure 9, how can the authors merge the separated immunoblots in to one panel? In addition of Red Ponceau, GAPDH or TUBULIN needs to be added as a loading internal control for densitometric normalization. The statistical results of p values of individual protein need to be shown in the bar graph. ­­­­­­­The total protein levels of HSBP1, HSP5A and VINC are distinct in leiomyoma and myometrium. It is interesting to know if that match with the changes of their phosphoprotein levels.

E.     Phosphorylation status of individual phosphoprotein should be validated by immunoblotting using serine, threonine, or tyrosine antibodies if applicable. Both immobilized metal affinity chromatography or Pro-Q Diamond staining are not completely specific that may produce false positive results.­­

Minors:

A.    There are numerous typographic and grammatic errors in the manuscript. For example, a space needs to be deleted between approach and were in Page 2 Line 36.

B.    Different numbers of differentially expressed phosphoproteins are showed in text, needs to be confirmed.

Author Response

Reviewer 1

1.      The protease inhibitors used in the sample preparation are major concerns. PMSF is a serine hydrolase inactivator and benzamidine is an inhibitor of trypsin, trypsin-like, and serine proteases. Inhibitors for cysteine protease, metalloprotease, calpain, and other proteases are not applied in the procedure. This may result in degradation of the phosphoproteins into small fragments that cannot be separated by 2-DE. Families of phosphoproteins or signal transduction pathways may be missed that could be physiologically important and alter the conclusion of the study.

Our reply: We thank the reviewer for this comment. We have chosen protease inhibitors (PMSF and benzamidine) that are routinely used in proteomic studies in addition to a mix of phosphatase inhibitors. Most importantly, however, the protein extraction solution (7 M urea, 2 M thiourea, 4% CHAPS, 40 mM Tris, 65 mM dithiothreitol (DTT) and 0.24% Bio-Lyte 3/10) is strongly denaturing and this leads to a marked decrease in protease activity, if not to a complete deactivation. As can be judged from figures 1A, 1B and 2, protein spots are clearly visible at all values of molecular weight, from very high (250 kDa) to very low (10 kDa) , thus supporting the conclusion that a possible residual activity of other proteases is minimal and does not affect the integrity of the study.

The authors compared the phosphoproteins between paired diseased and adjacent normal tissues from same patients. The change of abundance of protein spots is not so substantial, Ratios ≥ 1.5 and ≤ 0.6 are considered as significantly different. Comparison between diseased patients with “normal” patient controls may be more meaningful.

Our reply: In our study we applied stringent parameters for filtering the data from the 2-DE experiment. We used a fold change 1.5 and a Wilcoxon test (p-value<0.05) for the comparison of the phosphoproteins between paired diseased and adjacent normal tissues from same patients. A fold change ≥1.5 is indicated in the cases of the distinction between up or down proteins regulated for the reduction of the false positive rate and operator-related errors (Millioni R et al. Operator- and software-related post-experimental variability and source of error in 2-DE analysis. Amino Acids. 2012,42:1583-90) and it is widely accepted as cut-off value in proteomic studies. It is important to highlight that, differently from what it is often observed in transcriptomic and genomic studies, very high values of fold change are seldom retrieved in proteomics studies comparing samples that are similar, as in this case. In the proteomic literature a fold change of 1.5 is therefore normally accepted as demonstrated also in a recent study where ratios ≥ 1.5 and ≤ 0.6 were used for the identification of dysregulated proteins between the leiomyoma and the normal myometrium. (Jamaluddin MFB, Ko YA, Kumar M, et al. Proteomic Profiling of Human Uterine Fibroids Reveals Upregulation of the Extracellular Matrix Protein Periostin. Endocrinology. 2018 Feb 1;159(2):1106-1118. doi: 10.1210/en.2017-03018).

Regarding the suggestion of using normal patients as controls, not only it is difficult to obtain tissue from normal healthy subjects, but indeed the use of paired samples (normal and diseased) from the same patient strongly contributes to reduce the biological variability and helps to pinpoint real changes related to the pathology.

3. The biographic data of the patients are limited and clinic data are missing. The diagnosis is not clear. The histopathology of the diseased or adjacent tissues are not shown. The distance from diseased tissue to adjacent normal tissue is not described.

Our reply: All leiomyomas used in this study were confirmed histologically as benign ordinary leiomyomas. All leiomyomas selected for the study were subserosal/intramural, with dimension ranging between 4 to 6 cm.

Two samples were collected from each patient: one from the central area of the leiomyoma and one from the unaffected myometrium situated more than 2 cm away from the leiomyoma capsule. These details have now been added to the revised version of the manuscript.

4.      In Figure 9, how can the authors merge the separated immunoblots in to one panel? In addition of Red Ponceau, GAPDH or TUBULIN needs to be added as a loading internal control for densitometric normalization. The statistical results of p values of individual protein need to be shown in the bar graph. The total protein levels of HSBP1, HSP5A and VINC are distinct in leiomyoma and myometrium. It is interesting to know if that match with the changes of their phosphoprotein levels

Our reply. The three proteins validated in this study derive from the same blotting. The representation of Figure 9 is consistent with other studies (Jamaluddin MFB, Ko YA, Kumar M, et al. Proteomic Profiling of Human Uterine Fibroids Reveals Upregulation of the Extracellular Matrix Protein Periostin. Endocrinology. 2018 Feb 1;159(2):1106-1118. doi: 10.1210/en.2017-03018; Lin CP, Chen YW, Liu WH, et al. Proteomic identification of plasma biomarkers in uterine leiomyoma. Mol Biosyst. 2012 Apr;8(4):1136-45. doi: 10.1039/c2mb05453a).

To normalize the results of our WB analysis, we determined the total protein content of each sample by Red Ponceau. The reason for doing this is that the proteins that are usually selected as encoded by housekeeping genes (i.e. GAPDH or TUBULIN) are upregulated in leiomyoma and, thus, not adequate to be used as controls for normalization. (Ura B, Scrimin F, Arrigoni G, et al. A Proteomic Approach for the Identification of Up-Regulated Proteins Involved in the Metabolic Process of the Leiomyoma. Int J Mol Sci. 2016;17:540; Lemeer S, Gholami AM, Wu Z, Kuster B. Quantitative proteome profiling of human myoma and myometrium tissue reveals kinase expression signatures with potential for therapeutic intervention. Proteomics. 2015;15(2-3):356-64.)

Because we could not establish which proteins should be considered as housekeeping in our samples, we decided to apply a total protein content normalization method.

Following the reviewer’s suggestion, we added the statistical results of p values of individual proteins in the bar graph.

We also agree with the reviewer that it would be interesting to know if changes in abundance observed for these proteins match with changes of their phosphorylation levels. Unfortunately, specific anti-phospho sites antibodies for these proteins are not commercially available. Moreover, our approach does not allow us to identify the phosphorylation site(s) of the proteins, making it therefore difficult to confirm our results with specific antibodies.

 5.      Phosphorylation status of individual phosphoprotein should be validated by immunoblotting using serine, threonine, or tyrosine antibodies if applicable. Both immobilized metal affinity chromatography or Pro-Q Diamond staining are not completely specific that may produce false positive results.

Our reply: We thank the reviewer for this observation. As we have underlined above, the approach we chose for this study does not allow us to identify how many sites are phosphorylated in these proteins and which of them have an altered level in leiomyoma as compared to the normal tissue. The use of phosphosite-specific antibodies cannot be considered an option for validating our data. In our experience, commercially available antiphospho-Ser and antiphospho-Thr antibodies suffer from the same limitations as the IMAC and Diamond Pro-Q staining: they are not completely specific and may not give reliable results. On the other hand, antipohospho-Tyr antibodies are very much reliable and could be used as a validation method. However, considering that Tyr-phosphorylation is known to affect only a very small proportion of the global phosphoproteome (see data reported in the PhosphositePlus database at www.phosphositeplus.org) and that the level of Tyr phosphorylation is usually largely sub-stoichiometric, it is very unlikely that this kind of phosphorylation is responsible for the different abundance we have observed upon IMAC enrichment, between normal and diseased tissue.

Minors:

1.      There are numerous typographic and grammatic errors in the manuscript. For example, a space needs to be deleted between approach and were in Page 2 Line 36.

 Our reply: We improved the manuscript, by correcting typographic and grammatical errors.

 2.       Different numbers of differentially expressed phosphoproteins are showed in text, needs to be confirmed.

Our reply: We have now fixed the text.

Reviewer 1

1.      The protease inhibitors used in the sample preparation are major concerns. PMSF is a serine hydrolase inactivator and benzamidine is an inhibitor of trypsin, trypsin-like, and serine proteases. Inhibitors for cysteine protease, metalloprotease, calpain, and other proteases are not applied in the procedure. This may result in degradation of the phosphoproteins into small fragments that cannot be separated by 2-DE. Families of phosphoproteins or signal transduction pathways may be missed that could be physiologically important and alter the conclusion of the study.

Our reply: We thank the reviewer for this comment. We have chosen protease inhibitors (PMSF and benzamidine) that are routinely used in proteomic studies in addition to a mix of phosphatase inhibitors. Most importantly, however, the protein extraction solution (7 M urea, 2 M thiourea, 4% CHAPS, 40 mM Tris, 65 mM dithiothreitol (DTT) and 0.24% Bio-Lyte 3/10) is strongly denaturing and this leads to a marked decrease in protease activity, if not to a complete deactivation. As can be judged from figures 1A, 1B and 2, protein spots are clearly visible at all values of molecular weight, from very high (250 kDa) to very low (10 kDa) , thus supporting the conclusion that a possible residual activity of other proteases is minimal and does not affect the integrity of the study.

 The authors compared the phosphoproteins between paired diseased and adjacent normal tissues from same patients. The change of abundance of protein spots is not so substantial, Ratios ≥ 1.5 and ≤ 0.6 are considered as significantly different. Comparison between diseased patients with “normal” patient controls may be more meaningful.

 Our reply: In our study we applied stringent parameters for filtering the data from the 2-DE experiment. We used a fold change 1.5 and a Wilcoxon test (p-value<0.05) for the comparison of the phosphoproteins between paired diseased and adjacent normal tissues from same patients. A fold change ≥1.5 is indicated in the cases of the distinction between up or down proteins regulated for the reduction of the false positive rate and operator-related errors (Millioni R et al. Operator- and software-related post-experimental variability and source of error in 2-DE analysis. Amino Acids. 2012,42:1583-90) and it is widely accepted as cut-off value in proteomic studies. It is important to highlight that, differently from what it is often observed in transcriptomic and genomic studies, very high values of fold change are seldom retrieved in proteomics studies comparing samples that are similar, as in this case. In the proteomic literature a fold change of 1.5 is therefore normally accepted as demonstrated also in a recent study where ratios ≥ 1.5 and ≤ 0.6 were used for the identification of dysregulated proteins between the leiomyoma and the normal myometrium. (Jamaluddin MFB, Ko YA, Kumar M, et al. Proteomic Profiling of Human Uterine Fibroids Reveals Upregulation of the Extracellular Matrix Protein Periostin. Endocrinology. 2018 Feb 1;159(2):1106-1118. doi: 10.1210/en.2017-03018).

Regarding the suggestion of using normal patients as controls, not only it is difficult to obtain tissue from normal healthy subjects, but indeed the use of paired samples (normal and diseased) from the same patient strongly contributes to reduce the biological variability and helps to pinpoint real changes related to the pathology.

 3. The biographic data of the patients are limited and clinic data are missing. The diagnosis is not clear. The histopathology of the diseased or adjacent tissues are not shown. The distance from diseased tissue to adjacent normal tissue is not described.

 Our reply: All leiomyomas used in this study were confirmed histologically as benign ordinary leiomyomas. All leiomyomas selected for the study were subserosal/intramural, with dimension ranging between 4 to 6 cm.

Two samples were collected from each patient: one from the central area of the leiomyoma and one from the unaffected myometrium situated more than 2 cm away from the leiomyoma capsule. These details have now been added to the revised version of the manuscript.

4.      In Figure 9, how can the authors merge the separated immunoblots in to one panel? In addition of Red Ponceau, GAPDH or TUBULIN needs to be added as a loading internal control for densitometric normalization. The statistical results of p values of individual protein need to be shown in the bar graph. The total protein levels of HSBP1, HSP5A and VINC are distinct in leiomyoma and myometrium. It is interesting to know if that match with the changes of their phosphoprotein levels

 Our reply. The three proteins validated in this study derive from the same blotting. The representation of Figure 9 is consistent with other studies (Jamaluddin MFB, Ko YA, Kumar M, et al. Proteomic Profiling of Human Uterine Fibroids Reveals Upregulation of the Extracellular Matrix Protein Periostin. Endocrinology. 2018 Feb 1;159(2):1106-1118. doi: 10.1210/en.2017-03018; Lin CP, Chen YW, Liu WH, et al. Proteomic identification of plasma biomarkers in uterine leiomyoma. Mol Biosyst. 2012 Apr;8(4):1136-45. doi: 10.1039/c2mb05453a).

To normalize the results of our WB analysis, we determined the total protein content of each sample by Red Ponceau. The reason for doing this is that the proteins that are usually selected as encoded by housekeeping genes (i.e. GAPDH or TUBULIN) are upregulated in leiomyoma and, thus, not adequate to be used as controls for normalization. (Ura B, Scrimin F, Arrigoni G, et al. A Proteomic Approach for the Identification of Up-Regulated Proteins Involved in the Metabolic Process of the Leiomyoma. Int J Mol Sci. 2016;17:540; Lemeer S, Gholami AM, Wu Z, Kuster B. Quantitative proteome profiling of human myoma and myometrium tissue reveals kinase expression signatures with potential for therapeutic intervention. Proteomics. 2015;15(2-3):356-64.)

Because we could not establish which proteins should be considered as housekeeping in our samples, we decided to apply a total protein content normalization method.

Following the reviewer’s suggestion, we added the statistical results of p values of individual proteins in the bar graph.

We also agree with the reviewer that it would be interesting to know if changes in abundance observed for these proteins match with changes of their phosphorylation levels. Unfortunately, specific anti-phospho sites antibodies for these proteins are not commercially available. Moreover, our approach does not allow us to identify the phosphorylation site(s) of the proteins, making it therefore difficult to confirm our results with specific antibodies.

 5.      Phosphorylation status of individual phosphoprotein should be validated by immunoblotting using serine, threonine, or tyrosine antibodies if applicable. Both immobilized metal affinity chromatography or Pro-Q Diamond staining are not completely specific that may produce false positive results.

Our reply: We thank the reviewer for this observation. As we have underlined above, the approach we chose for this study does not allow us to identify how many sites are phosphorylated in these proteins and which of them have an altered level in leiomyoma as compared to the normal tissue. The use of phosphosite-specific antibodies cannot be considered an option for validating our data. In our experience, commercially available antiphospho-Ser and antiphospho-Thr antibodies suffer from the same limitations as the IMAC and Diamond Pro-Q staining: they are not completely specific and may not give reliable results. On the other hand, antipohospho-Tyr antibodies are very much reliable and could be used as a validation method. However, considering that Tyr-phosphorylation is known to affect only a very small proportion of the global phosphoproteome (see data reported in the PhosphositePlus database at www.phosphositeplus.org) and that the level of Tyr phosphorylation is usually largely sub-stoichiometric, it is very unlikely that this kind of phosphorylation is responsible for the different abundance we have observed upon IMAC enrichment, between normal and diseased tissue.

 Minors:

1.      There are numerous typographic and grammatic errors in the manuscript. For example, a space needs to be deleted between approach and were in Page 2 Line 36.

 Our reply: We improved the manuscript, by correcting typographic and grammatical errors.

 2.       Different numbers of differentially expressed phosphoproteins are showed in text, needs to be confirmed.

 Our reply: We have now fixed the text.

Reviewer 2 Report

The manuscript “Phosphoproteins involved in the tumorigenic signaling pathway, in the inhibition of apoptosis and in cell survival, identified through the phosphoproteomic analysis of the leiomyoma” submitted by Blendi Ura et al. provides an exhaustive characterization of the role of phosphoproteins in leiomyoma. In particular, the authors applied a comparative phosphoproteomic analysis to identify a set of phosphoproteins up or downregulated in leiomyoma in comparison with control myometrium.  Overall, the paper supplies a wealth of additional information on the phosphoproteome in leiomyoma that can be interesting for the field. However, the work should be completed with further experiments to demonstrate the role of the phosphoproteins identified. I strongly recommend the incorporation of additional data to improve the final quality of the work.

 Major comments:

 1)     Lines 64-67- Please provide more information in this section regarding the role of kinases in leiomyomas. Different kinase inhibitors have been demonstrated effective against leiomyoma growth. Please provide information regarding the state of the art of such inhibitors.  

 2)     Among all the phosphoproteins identified, UBA1 was identified twice with a fold change of 3.8 and 2.7. This protein catalyzes the first step in ubiquitination, an essential step to mark proteins for its degradation.  Due to its central role in ubiquitination, UBA1 has been linked to cell cycle regulation, endocytosis, signal transduction, apoptosis, DNA damage repair, transcriptional regulation and cancer. The authors should discuss the role of this protein and ubiquitination in the development of leiomyoma. Also, I would suggest the authors the validation of the phosphorylation of this protein and study the role of this modification in the development and progression of leiomyoma.

 3) Line 321-328- Please explain and discuss the criteria followed for the selection of HSPB1, HSP5A and VINC.

 4) The effects of specific kinase inhibitors should be experimentally demonstrated in order to correlate their effects with changes in the phosphorylation status of the identified proteins.

 5)  Line 416-423- Please discuss the role of kinase inhibitors in leiomyoma.  

 6) I would recommend the reorganization and simplification of the current title of the manuscript.

Author Response

Reviewer 2

 1)            Lines 64-67- Please provide more information in this section regarding the role of kinases in leiomyomas. Different kinase inhibitors have been demonstrated effective against leiomyoma growth. Please provide information regarding the state of the art of such inhibitors.

 Our reply: We added: Several studies have shown the dysregulation of the PI3K/Akt/mTOR pathway. Indeed, they find an increase in cyclin D2 and glycogen synthase kinase 3 (GSK3) and decrease of PTEN leading to leiomyoma development (13). There is an evidence that the Ras/Raf/MEK/ERK pathway is dysregulated in leiomyoma. Several studies have shown that different kinases, such as ERK and RTK, are altered in this pathway and this may be related to the disease (14,15).

Information regarding specific inhibitors affecting leiomyoma growth has been added to the discussion section.

 2)            Among all the phosphoproteins identified, UBA1 was identified twice with a fold change of 3.8 and 2.7. This protein catalyzes the first step in ubiquitination, an essential step to mark proteins for its degradation. Due to its central role in ubiquitination, UBA1 has been linked to cell cycle regulation, endocytosis, signal transduction, apoptosis, DNA damage repair, transcriptional regulation and cancer. The authors should discuss the role of this protein and ubiquitination in the development of leiomyoma. Also, I would suggest the authors the validation of the phosphorylation of this protein and study the role of this modification in the development and progression of leiomyoma.

 Our reply: We thank the reviewer for this useful comment. The UBA1 is an important enzyme whose activity is related to cell cycle regulation, endocytosis, apoptosis and cancer. The ubiquitination is a poorly studied process in leiomyoma, although it is known that the blocking of this process is related to the growth of tumors. An example is the switching of the ubiquitin/proteasome-dependent degradation of RXRα by phosphorylation in leiomyomas. These events may be responsible for the accumulation of RXRα and the consequent dysregulation of retinoic acid target genes (53).

UBA1 catalyzes the first step in ubiquitin conjugation to mark cellular proteins for degradation through the ubiquitin-proteasome system (54). This enzyme role has been linked to DNA repair, for response to replication stress, cell cycle regulation, apoptosis and cancer (55). UBA1 inhibitors lead to an unfolded protein response and induces cell death in malignant cells over normal cells (56).

As already discussed in the answer to the point raised by Reviewer 1 (see above) our approach does not allow to identify the specific phosphorylation sites present in these proteins. Therefore, a validation with specific antiphosphosite antibodies cannot be carried out. For further details regarding the use of generic antibodies against phosphor-Ser, -Thr or -Tyr, please refer to what we have reported above.

We appreciate the suggestion of the reviewer regarding the possibility of further studying the phosphorylation of UBA1 in relation to leiomyoma and we might start a new research line following this suggestion.

 3) Line 321-328- Please explain and discuss the criteria followed for the selection of HSPB1, HSP5A and VINC.

 Our reply: We selected HSPB1 and HSP5A because, based on our bioinformatic analysis, these two proteins are associated to the inhibition of the apoptosis and cell proliferation. We selected VINC because the antibody is commercially available.

 4) The effects of specific kinase inhibitors should be experimentally demonstrated in order to correlate their effects with changes in the phosphorylation status of the identified proteins.

 Our reply: We agree with the reviewer that it would be important to verify the altered phosphorylation level of the proteins with the use of specific kinase inhibitors. However, this would be relatively easy using an in vitro model (i.e. cell lines or primary cell cultures treated with specific inhibitors) but it is not doable in our case, since we have performed our study starting from human tissues. Moreover, as we have already discussed above (see reply to Reviewer 1), our approach does not allow to identify the phosphorylation site(s) of the proteins. Without this information, it is hard to guess which kinase(s) might be responsible for such phosphorylation and select the appropriate inhibitors to be tested.

 Line 416-423- Please discuss the role of kinase inhibitors in leiomyoma.

 Our reply: As suggested by the reviewer, we have now added a comment regarding the role of kinase inhibitors in the leiomyoma: “Several studies have been conducted to highlight the effects of various kinase inhibitors in the leiomyoma. Shushan A et al. (51) used the AG1478 an EGFR kinase blocker in leiomyoma cells. Leiomyoma cell growth is inhibited by AG1478, and is unaffected by the presence of physiological concentrations of estradiol and progesterone. In another paper, Shushan A et al. (52) evaluate the efficiency of genistein (a plant flavonoid) and the new protein tyrosine kinase inhibitor TKS050 in the inhibition of autophosphorylation of EGFR and downstream signal transduction events, including cell proliferation and cell cycle progression.

 I would recommend the reorganization and simplification of the current title of the manuscript.

 Our reply: We simplified the current title of the manuscript in: Phosphoproteins involved in the inhibition of apoptosis and in cell survival in the leiomyoma.

Reviewer 3 Report

The authors present an interesting article regarding the differential expression or phosphorylation of 32 proteins in the leiomyoma when compared to the endometrium. A possible drawback of the present manuscript is the small number of samples retrieved.

The overall quality of the manuscript could be improved by minor alterations throughout the text. i.e. lines 78-79 are common knowledge for the readership, thus these lines should be removed, especially since they are followed by the aim of the study. Another example is in line 366 “nothing is known” should be replaced by “no data regarding… have been reported yet”.

Some (if not all) figures 3-8 should be reported as supplementary material as they do not present enough interest in the results section.

In lines 421-423 authors state that ‘’In our opinion, our data shed light on mechanisms that still need to be better understood, but that could open the path to the development of a new class of drugs that could not only block the development, but could also lead to a significant reduction in tumor size’’. Please specify the possible molecular mechanisms via which the proposed drugs will affect the tumor development, while considering that the pathways indicated to be associated with tumor development are main regulators of the normal cell cycle.

Finally, in the discussion section the authors fail to include the most recent literature about protein expression in leiomyoma.

Looking forward to reading the revised version of the manuscript

 Author Response

Reviewer 3

1.      The authors present an interesting article regarding the differential expression or phosphorylation of 32 proteins in the leiomyoma when compared to the endometrium. A possible drawback of the present manuscript is the small number of samples retrieved.

 Our reply: We thank the reviewer for this observation. In proteomic discovery and verification studies sample are generally small (max 10) for reasons of cost, time needed for the analysis and data processing (Rifai N, Gillette MA, Carr SA. Protein biomarker discovery and validation: the long and uncertain path to clinical utility. Nat Biotechnol. 2006; 24:971–983). The number of patients used in our work is in line with other studies:

1.      Liu Y, Lu D, Sheng J, Luo L, Zhang W. Identification of TRADD as a potential biomarker in human uterine leiomyoma through iTRAQ based proteomic profiling. Mol Cell Probes. 2017 Dec;36:15-20. doi: 10.1016/j.mcp.2017.07.001.

2.      Lemeer S, Gholami AM, Wu Z, Kuster B. Quantitative proteome profiling of human myoma and myometrium tissue reveals kinase expression signatures with potential for therapeutic intervention. Proteomics. 2015 Jan;15(2-3):356-64. doi: 10.1002/pmic.201400213.

3.      Lin CP1, Chen YW, Liu WH, et al. Proteomic identification of plasma biomarkers in uterine leiomyoma. Mol Biosyst. 2012 Apr;8(4):1136-45. doi: 10.1039/c2mb05453a.

 2.      The overall quality of the manuscript could be improved by minor alterations throughout the text. i.e. lines 78-79 are common knowledge for the readership, thus these lines should be removed, especially since they are followed by the aim of the study. Another example is in line 366 “nothing is known” should be replaced by “no data regarding… have been reported yet”.

Our reply: We thank the reviewer for his comment. We have removed the following sentence: “Two-dimensional gel electrophoresis (2-DE) is a powerful method for the quantification of thousands of proteins, including isoform or protein post-translational modifications (PTMs).” Line 366 has been improved according to the indications of the reviewer.

3.      Some (if not all) figures 3-8 should be reported as supplementary material as they do not present enough interest in the results section.

Our reply: Figures 3-8 have been moved to the supplementary material.

4.      In lines 421-423 authors state that ‘’In our opinion, our data shed light on mechanisms that still need to be better understood, but that could open the path to the development of a new class of drugs that could not only block the development, but could also lead to a significant reduction in tumor size’’. Please specify the possible molecular mechanisms via which the proposed drugs will affect the tumor development, while considering that the pathways indicated to be associated with tumor development are main regulators of the normal cell cycle.

Our reply: We have rewritten the sentence: In our opinion, our data provided represent a step forward in the difficult understanding of the molecular mechanisms that lead to the formation and development of leiomyoma.

5.      Finally, in the discussion section the authors fail to include the most recent literature about protein expression in leiomyoma.

Our reply: We have now added a part of the discussion regarding the most recent literature related to protein expression changes associated to leiomyoma: “Jamaluddin MFB et al. (14) conducted a proteomic study for the characterization of fibroid ECM proteins. They identified several proteins up regulated in the leiomyoma: POSTN, TNC, COL3A, COL24A, and ASPN. Another proteomic study conducted by Liu Y et al. (24) identified TRADD as a potential biomarker in human uterine leiomyoma.

Reviewer 4 Report

The authors describe a phosphoproteomic analysis by Pro-Q and IMAC to identify possible protein dysregulation underlying leiomyomas. A few points to consider.

-The introduction needs to be improved for flow. As it stands it is a description of a few protein modifications that have been associated with leiomyomas. There needs to be connection between the paragraphs and a better description of what you are studying and why. There is some good information there, it just needs to be improved for readability.

- did you use some method to control for false discovery (particular fold change and p-value), FDR cutoff? Please state this your statistical methods section

-what were the IPA statistical cutoffs you used for reporting? Or just top 2 pathways?

-when you refer to abundance are you referring to protein abundance or phosphorylation level? Did you control for protein abundance in your phosphorylation analyses since this could influence results (e.g., higher protein abundance could lead to higher phosphorylation just due to total amount of protein detected).

-which proteins do the numbers in the gel figures correlate with?

-the discussion is good in that it brings in your major pathways (apoptosis and cell survival) and some supporting literature. Could you possible briefly discuss your panther analysis and the implications there? (e.g., protein class, cellular component, etc).

Author Response

Reviewer 4

 1.      The introduction needs to be improved for flow. As it stands it is a description of a few protein modifications that have been associated with leiomyomas. There needs to be connection between the paragraphs and a better description of what you are studying and why. There is some good information there, it just needs to be improved for readability.

Our reply: Following the reviewer’s suggestion, we have now changed the introduction and we hope that it is now more readable.

2.      Did you use some method to control for false discovery (particular fold change and p-value), FDR cutoff? Please state this your statistical methods section.

Our reply: We thank the reviewer for this comment. We did not apply any correction to our results to adjust for multiple comparisons.

3.      What were the IPA statistical cutoffs you used for reporting? Or just top 2 pathways?

Our reply: In IPA, we considered P<0.05 as a statistically significant value.This is now specified in the text.

4.      When you refer to abundance are you referring to protein abundance or phosphorylation level? Did you control for protein abundance in your phosphorylation analyses since this could influence results (e.g., higher protein abundance could lead to higher phosphorylation just due to total amount of protein detected).

Our reply: We thank the reviewer for this comment. As we stated in the discussion section, we always refer to the abundance of putative phosphoproteins and not to the actual phosphorylation level. Indeed, as correctly pointed out by the reviewer, an altered abundance in protein spot could be related either to an altered phosphorylation level or to an altered protein abundance, or both. In our experiments conducted with the Diamond Pro-Q dye, we have also stained the gel with sypro ruby to verify that the total protein content was the same (Fig 1B). For the IMAC experiments the starting material was quantified in the same way, therefore we are quite confident that no significant different amounts of proteins were loaded into our IMAC columns. The conduct of experiments, data processing and evaluations are in line with the literature.

1.      Khang R, Park C, Shin JH. The biguanide metformin alters phosphoproteomic profiling in mouse brain. Neurosci Lett. 2014 Sep 5;579:145-50.

2.      Forthun RB, Aasebø E, Rasinger JD, Bedringaas SL, Berven F, Selheim F, Bruserud Ø, Gjertsen BT. Phosphoprotein DIGE profiles reflect blast differentiation, cytogenetic risk stratification, FLT3/NPM1 mutations and therapy response in acute myeloid leukaemia. J Proteomics. 2018;173:32-41.

5.      Which proteins do the numbers in the gel figures correlate with?

Our reply: The numbers of the proteins in the gel figures coincide with the number of proteins reported to the Table 1.

6.      The discussion is good in that it brings in your major pathways (apoptosis and cell survival) and some supporting literature. Could you possible briefly discuss your panther analysis and the implications there? (e.g., protein class, cellular component, etc).

Our reply: We added to the discussion: “Panther analysis revealed a significant enrichment of GO terms for several important processes, like metabolic processes and cellular biogenesis. Some of these metabolic processes have been previously described by our group (16), pointing to a particular importance of metabolism in leiomyoma growth. In addition, our Panther analysis reveals that many proteins identified in our study are enzymes, are related to oxidative stress, and reside mostly inside the cell.

Round  2

Reviewer 1 Report

The manuscript has been improved. However, the merged blots in Figure 9 have not been coped, should be corrected for publication.

Author Response

Reviewer 1

The manuscript has been improved. However, the merged blots in Figure 9 have not been coped, should be corrected for publication.

Our reply: We thanks the reviewer for the suggestion. We send the figure 9 improved.

Reviewer 2 Report

After revision, the authors improved the quality of the work by including new discussions and clarifying some of the drawbacks present in the previous version of the manuscript. I recommend the publication of the manuscript in the present form.

Author Response

We thank the reviewer for his suggestions that improved the manuscript.

Reviewer 4 Report

The authors have done a good job revising their manuscript according to all reviewer comments. I have no further comments for improvement.

Author Response

We thank the reviewer for his suggestions that improved the manuscript